# Highly efficient *in crystallo* energy transduction of light to work

Jiawei Lin[1,2], Jianmin Zhou[1,2], Liang Li[3,4], Ibrahim Tahir [3], Songgu Wu [1,2] ✉, Pan*e Naumov [3,5,6,7] ✉ & Junbo Gong [1,2] ✉

Various mechanical effects have been reported with molecular materials, yet organic crystals capable of multiple dynamic effects are rare, and at present, their performance is worse than some of the common actuators. Here, we report a confluence of different mechanical effects across three polymorphs of an organic crystal that can efficiently convert light into work. Upon photo-dimerization, acicular crystals of polymorph I display output work densities of about 0.06–3.94 kJ m$^{-3}$, comparable to ceramic piezoelectric actuators. Prismatic crystals of the same form exhibit very high work densities of about 1.5–28.5 kJ m$^{-3}$, values that are comparable to thermal actuators. Moreover, while crystals of polymorph II roll under the same conditions, crystals of polymorph III are not photochemically reactive; however, they are mechanically flexible. The results demonstrate that multiple and possibly combined mechanical effects can be anticipated even for a simple organic crystal.

Dynamic molecular crystals are an emerging researched class of engineering materials that are thought to hold great promise as responsive, lightweight, flexible, and potentially biocompatible energy-converting materials[1–3]. They are known to exhibit remarkable dynamic effects of restorative or disintegrative nature that can include bending[4–7], twisting[8–10], jumping[11,12], fracturing[13], shape-memory[14], and self-healing[15–18], some of which hold potential for use in actuators[19–21], flexible electronics[22–25], optical waveguides[26–29], and sensors[30,31], among a plethora of other prospective applications. The process of energy transduction that these materials are usually capable of involves the input of energy such as heat, light, or mechanical force and output in the form of kinetic energy that can be used to perform mechanical work—a process whose energy cost relies directly on the amplification of the molecular-scale changes for reshaping or autonomous propulsion on a macroscopic level. Beyond the apparent advantage of having low densities and therefore being light in weight, the organic crystalline materials are amenable to chemical alteration of their physicochemical properties and span significantly large temperature ranges

(for an organic material) of mechanical stability. This property caters to their future applications at high or low temperatures.

Driven by these prospective applications, most of the current research efforts are focused on gaining a deeper understanding of the role of intermolecular interactions and the establishment of predictive approaches toward their physicochemical properties[1,9,32]. At the outset, the mechanically flexible crystals are typically categorized as plastic or elastic, depending on their response and propensity for shape recovery after applying mechanical force[33,34]. Despite the significant efforts being made, currently, a reliable design of mechanical properties such as flexibility is generally not possible for molecular crystals since the role of molecular movement and intermolecular interactions to dissipate strain are intricately related to the crystal packing[32,35,36]. When compared to mechanical or temperature-driven control over the macroscopic shape, actuation by light comes with advantages, most notably, the possibility for remote control and precise input of energy. Light-triggered motions or deformations of molecular crystals have been established for several photochemical reactions, as has been

[1]School of Chemical Engineering and Technology, State Key Laboratory of Chemical Engineering, Tianjin University, Tianjin 300072, China. [2]Haihe Laboratory of Sustainable Chemical Transformations, Tianjin 300192, China. [3]Smart Materials Lab, New York University Abu Dhabi, PO Box, 129188 Abu Dhabi, UAE. [4]Department of Sciences and Engineering, Sorbonne University Abu Dhabi, PO Box, 38044 Abu Dhabi, UAE. [5]Center for Smart Engineering Materials, New York University Abu Dhabi, PO Box, 129188 Abu Dhabi, UAE. [6]Research Center for Environment and Materials, Macedonian Academy of Sciences and Arts, Bul. Krste Misirkov 2, MK–1000 Skopje, Macedonia. [7]Molecular Design Institute, Department of Chemistry, New York University, 100 Washington Square East, New York, NY 10003, USA. ✉e-mail: wusonggu@tju.edu.cn; pance.naumov@nyu.edu; junbo_gong@tju.edu.cn

demonstrated by *trans–cis* photoisomerization[37,38], photocyclization[39,40], and photocycloaddition[41,42].

However, two of the greatest challenges in this line of pursuit are low efficiency and lack of multifunctionality—highly desired yet difficult goals to pursue. The actuation efficiency is hard to predict, and for small objects like organic crystals, it is also practically difficult to measure. On the other hand, achieving multiple mechanical motions from the same compound is particularly challenging due to the difficulties in predicting the crystal packing, intermolecular interactions, and crystal habits. Here we use the well-established dimerization reaction that has been used ubiquitously in solid-state chemistry research due to its reliable relation with geometric criteria[43] to address both of these challenges and explore the dependence of multiple mechanical effects on the crystal packing of (Z)−2-(3,5-bis(trifluoromethyl)phenyl)−3-(1-methyl-1H-imidazol-2-yl) acrylonitrile (PMA). Three polymorphs of this material (PMA-I, PMA-II, and PMA-III) were obtained under different conditions and exhibited remarkably different mechanical effects. Upon exposure to UV irradiation, crystals of PMA-I display various photosalient effects, including splitting, jumping, and bending, depending on their respective habits and the conditions of excitation. On the contrary, crystals of PMA-II roll upon UV irradiation. Crystals of PMA-III do not exhibit photosalient behavior; however, they are mechanically flexible. To put this material into perspective of applications, the performance of one of the forms (form I) was quantified. The results showed a good performance against the commonly available actuators. The results indicate that the multifunctionality in dynamic crystalline materials is a combination of factors that probably involve mechanical and structural contributions.

## Results and discussion

PMA was synthesized by Knoevenagel condensation between 3,5-bis(trifluoromethyl)phenylacetonitrile and 1-methyl-1H-imidazole-2-carbaldehyde (Fig. 1a, Supplementary Fig. 1). The product was confirmed by NMR spectroscopy (Supplementary Fig. 2). Acicular crystals were obtained by layering ethanol on top of a dichloromethane solution of PMA and allowing for solvent diffusion at room temperature. Depending on the crystallization conditions, single crystals of different polymorphs with different colors and fluorescence were obtained (Fig. 1b–d). Green crystals with blue-green fluorescence of polymorph PMA-I crystallized at room temperature as two habits, acicular (needle-like) and prismatic (blocky). A larger amount of prismatic PMA-I crystals were prepared by evaporative crystallization from a solution in a mixture of dichloromethane and ethanol. Blue-fluorescent crystals of polymorph PMA-III were readily obtained at 5 °C. Colorless crystals of polymorph PMA-II crystallized concomitantly with crystals of PMA-I. However, their number was very small. The different forms can be readily distinguished by fluorescence spectroscopy, where the emission peaks of PMA-I, PMA-II, and PMA-III appear at 449, 474, and 464 nm, respectively (Fig. 2b). Single-crystal X-ray diffraction analysis confirmed the identity of the crystals of different polymorphs (Supplementary Table 1).

The chemical structure of PMA implies that the molecule is amenable to [2 + 2] photocycloaddition (Fig. 1a). For this reaction, it has been established that the intermolecular geometry affects the reaction path and, therefore, the strain tensor[5]. Since it has been postulated that the photochemical effects are related to the crystal size/shape and irradiation conditions[44], we investigated the photoactuation properties of crystals of the polymorphs of PMA having different crystal habits. As shown in Supplementary Movie 1, upon exposure to ultraviolet radiation, prismatic crystals of PMA-I showed remarkable photosalient effects and underwent disintegration and/or jumping, which was accompanied by fluorescence quenching (Fig. 1e). Face indexing showed that these crystals of PMA-I have three dominant facets, with the widest facet corresponding to (001)

(Supplementary Fig. 3a). The crystal disintegration tends to occur along the [010] direction. High-speed camera recordings of the phenomenon revealed that the photosalient effects occur within 20 ms (Supplementary Fig. 4, Supplementary Movie 2), showing a fast process. The disintegration of the crystals during a photosalient effect has been related to the sudden release of energy, possibly as a result of a phase transition coupled with the photochemical reaction[45,46].

In contrast with the prismatic crystals of PMA-I, when acicular crystals of PMA-I were irradiated, they bent away from the light source (Fig. 1f, Supplementary Fig. 5, Supplementary Movie 3). When the bending strain exceeded the elastic limit, they fragmented with salient behavior. By controlling the direction of the incident ultraviolet radiation, the crystals can be bent both perpendicular to (001) and (100) facets (Supplementary Fig. 5). The deformation is localized and plastic; by sequential focusing the excitation light on different crystal locations, a crystal with an undulated shape was obtained (Fig. 1g). The bending direction, away from the light source, can be attributed to the expansion of the (partially) reacted crystal surface. Contrary to PMA-I, however, the acicular crystals of PMA-II responded without obvious deformation and disintegration; instead, they rolled within 10 s of exposure to UV light (Fig. 1h, Supplementary Movie 4). In stark contrast with PMA-I and PMA-II, PMA-III did not show any observable macroscopic mechanical response.

The crystal structures of three forms were determined to gain insight into the molecular packing arrangement and intermolecular interactions of the three polymorphs (Supplementary Table 1). PMA-I crystallizes in the orthorhombic system ($a$ = 7.9407(3) Å, $b$ = 12.8084(5) Å, and $c$ = 28.4658 (9) Å), crystals of PMA-II are monoclinic ($a$ = 15.4734(9) Å, $b$ = 7.8129(3) Å, $c$ = 13.1933(6) Å, $\beta$ = 114.316(6)°), and crystals of PMA-III are monoclinic ($a$ = 17.5836(3) Å, $b$ = 9.2458(2) Å, $c$ = 9.1922(2) Å, $\beta$ = 93.653(2)°). Similar to other cases[43], the photomechanical effects were found to depend on the molecular disposition within the π-dimers (Fig. 3a, b, Supplementary Table 2). The molecules in the structures of the three polymorphs exhibit different conformations (Supplementary Fig. 6). We further analyzed the crystal packing and intermolecular interactions to identify the structural origin of different photochemical behavior. In accordance with Schmidt's topochemical criteria, the active dimers tend to react when the distance of parallel C=C bonds is less than 4.2 Å. In PMA-I, the two molecules of π dimers are arranged in an antiparallel and head-to-tail manner (Fig. 3a). The C=C bonds are perfectly aligned with $\theta_1$ = 0.0°, $\theta_2$ = 95.7° and $\theta_3$ = 56.3°, and the distance between two olefin bonds is 4.045 Å, and thus shorter than the approximate threshold for reactivity of 4.2 Å. Furthermore, the π-dimers connect with the neighboring molecules via C−H···N ($D$, $d$, and $\theta$ are 3.476 Å, 2.609 Å, and 151.9°; 3.426 Å, 2.615 Å, and 140.2°) and C−H···Cl (3.835 Å, 2.631 Å, 153.0°). In PMA-II, the two C=C bonds of the photoreactive dimers are also arranged in an antiparallel and head-to-tail manner and are parallel with each other, with $\theta_1$ = 0°, $\theta_2$ = 90.6° and $\theta_3$ = 67.6° (Fig. 3b), and the distance between them is 3.726 Å, that is, shorter than in PMA-I. The molecules in PMA-II are assembled via a more extensive hydrogen bonding network composed of C−H···N (3.405 Å, 2.720 Å, and 127.3°; 3.259 Å, 2.711 Å, and 115.8°; 3.525 Å, 2.594 Å, and 166.3°; 3.429 Å, 2.518 Å, and 160.8°) and C−H···Cl (3.352 Å, 2.521 Å, 146.17°) interactions (Fig. 3c, d). The C=C bonds in the two types of active dimers are perfectly parallel, and the distances between them are shorter than 4.2 Å. However, parallel dimers are not found in the crystal packing of PMA-III. The two C=C bonds are 4.972 Å apart, and the inter-dimer angles are $\theta_1$ = 42.0°, $\theta_2$ = 103.6° and $\theta_3$ = 46.5°, which explains the lack of reactivity of this polymorph (Supplementary Fig. 7).

Crystals of the reactive polymorphs PMA-I and PMA-II were irradiated for 1 h with ultraviolet light (maximum at 365 nm). The irradiated crystals were dissolved in DMSO-d$_6$ for ¹H NMR spectroscopic analysis, and the spectra of the products were practically identical (Fig. 2a, Supplementary Fig. 8). Compared to the ¹H NMR spectrum of

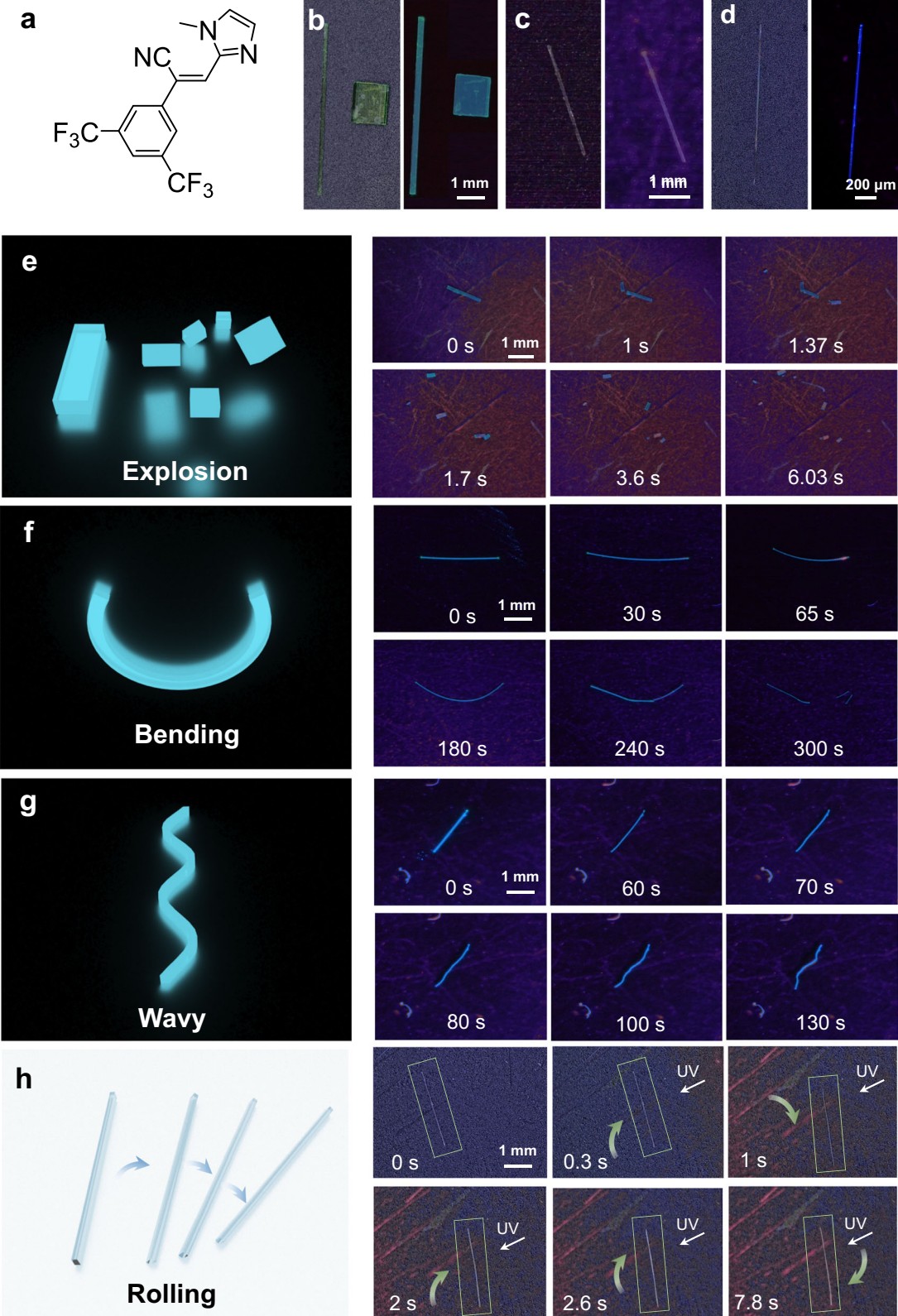

**Fig. 1 | Appearance and mechanical effects of crystals of the PMA polymorphs. a** Chemical formula of PMA. **b–d** Images of crystals of the three polymorphs of PMA under visible and ultraviolet light: PMA-I (**b**), PMA-II (**c**), and PMA-III (**d**). **e–h** Photomechanical effects of PMA-I (**e–g**) and PMA-II (**h**) exposed to ultraviolet light.

the sample before irradiation, the signal from the proton at the double bond (8.07 ppm) disappeared and a new signal at 6.04 ppm appeared, showing the generation of the cyclobutane ring. The three signals from the aromatic protons shifted from 8.43, 7.51 and 7.30 ppm to 7.82, 7.25 and 6.83 ppm, respectively, and one signal at 8.15 ppm remained unshifted. The signals from the methyl group shifted from 3.94 to 3.91 ppm (Supplementary Figs. 2 and 8). The photoproducts of PMA-I and PMA-II had different X-ray diffraction patterns that were also different

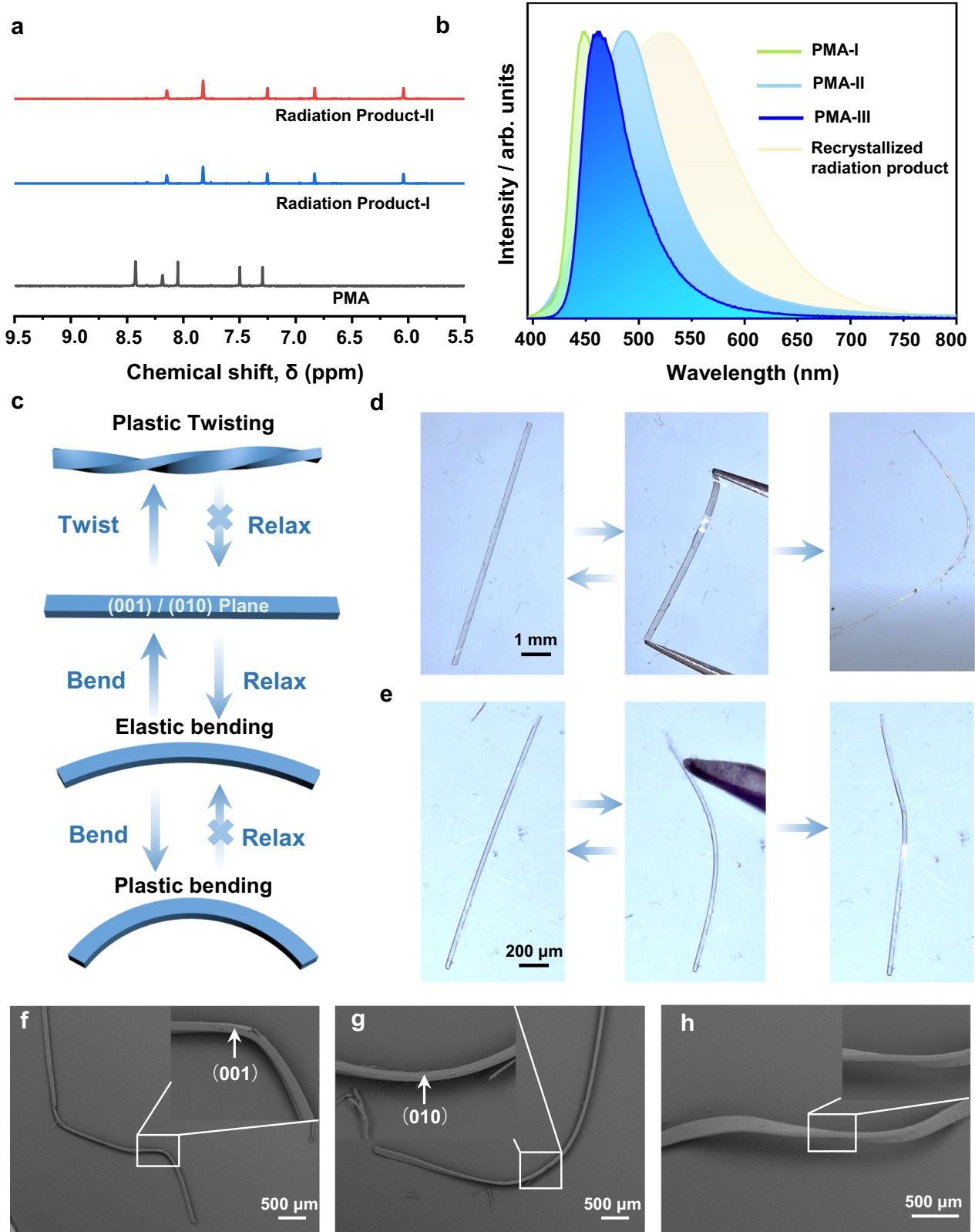

**Fig. 2 | Spectroscopic and mechanical characterization. a** ¹HNMR spectra of non-irradiated PMA and irradiated samples of PMA-I and PMA-II. **b** Fluorescence spectra of three forms and the photoreaction product. **c** Schematic diagram of the mechanical properties of PMA-III. **d**–**e** Micrographs of PMA-III crystals that were bent along their (001) and (010) planes. **f**–**h** Scanning electron micrographs of bent and twisted PMA-III crystals.

from that of the recrystallized product, and therefore they were different phases (Supplementary Fig. 9). A high-quality single crystal was prepared by recrystallizing the product from acetonitrile. The fluorescence spectrum of the recrystallized cyclobutane product has a characteristic peak at 541 nm that was not present in the spectra of the reactants (Fig. 2b). A crystal of the product was analyzed by single crystal X-ray diffraction analysis to determine its structure (Supplementary Table 1, Supplementary Fig. 10). The photoproduct

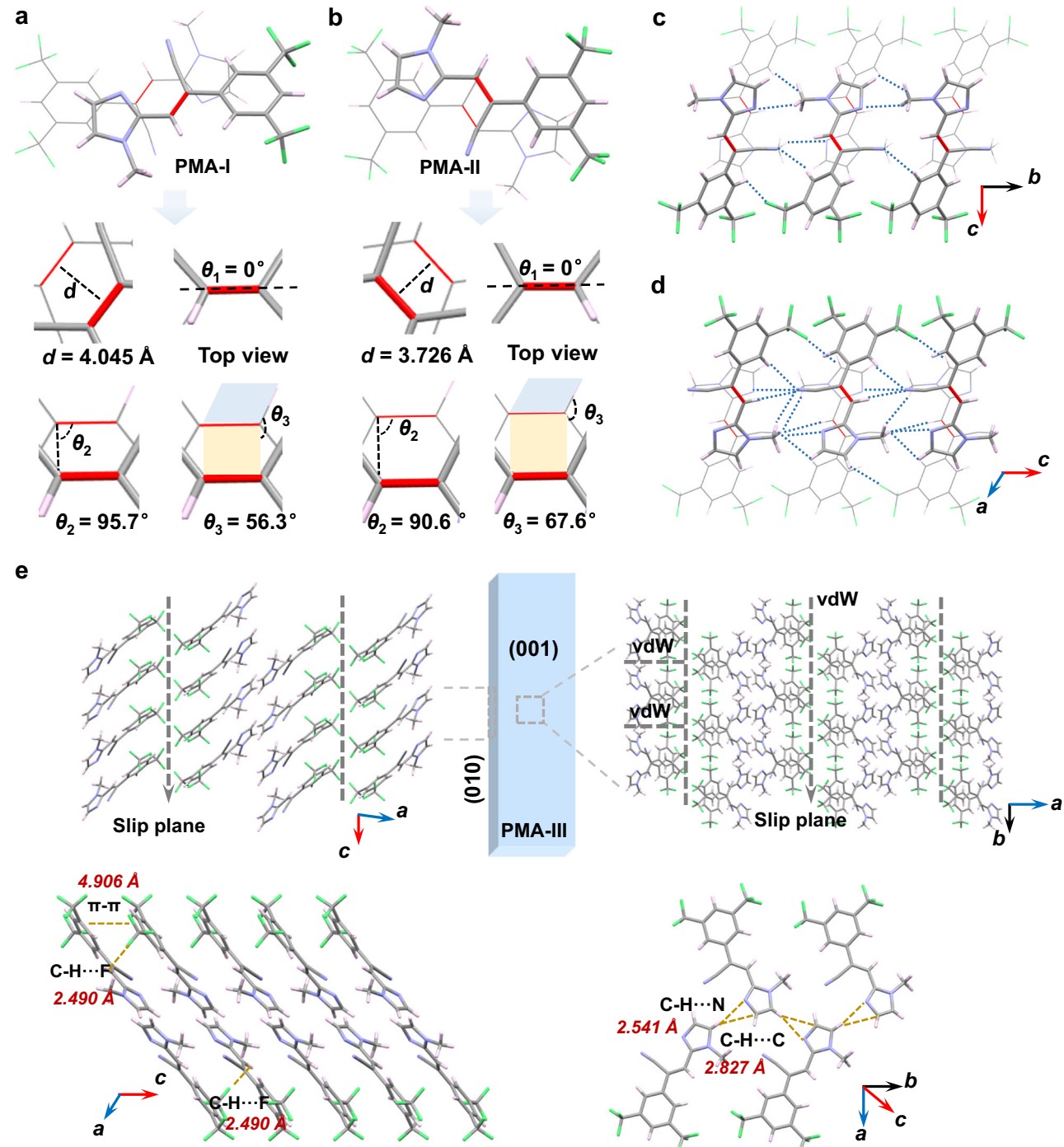

**Fig. 3 | Crystal structures of the three forms of PMA. a** Intra-π-dimer structure of PMA-I. The yellow plane represents the plane defined by the two C−C double bonds. The blue plane represents the plane defined by the C−C double bond and its adjacent single bond. **b** Intra-π-dimer structure of PMA-II. **c** Inter-π-dimer packing of PMA-I. **d** Inter-π-dimer packing of PMA-II (**e**) Molecular packing of PMA-III. The distances of the most relevant intermolecular interactions are highlighted in red and the red lines represent double bonds.

crystallizes in the monoclinic system ($a = 10.0905(4)$ Å, $b = 24.4237(8)$ Å, $c = 12.9626(5)$ Å and $\beta = 107.872(4)°$). The unit cell volume of the product is 3040.44 Å$^3$, and therefore nearly twice the volume of PMA-II (1453.47 Å$^3$). When ultraviolet light is shone perpendicular to the (100) or (001) planes, the radiation intensity decreases throughout the crystal due to the gradient shielding effect of the molecules[5]. The proximity to the light source promotes higher reaction yield and enhances the formation of the photodimerization product. The expansion of the unit cell ensuing from the photodimerization

generates stress, leading to bending of the acicular crystals of PMA-I away from the light. According to the Euler–Bernoulli equation[47,48], thick crystals with larger thickness will experience more strain when reaching the same radius of curvature, which, since they cannot bend, can concentrate at defects and result in crystal explosion. This has already been demonstrated on photosalient crystals undergoing linkage isomerization[49] as well as with thermally induced disintegrative effects, known as the thermosalient effect[11]. We hypothesize that the rolling behavior observed with PMA-II may arise from mechanical

instability caused by the dynamic structure conversion between PMA-II and the radiation product, resulting in strain release[50]. Further investigation into the relationship between macroscopic photomechanical motion and crystal structure may require mechanical simulations to explain thoroughly the complex kinematic effects.

As mentioned above, mechanically flexible crystals can be categorized as elastic and plastic. However, as with any object, each crystal comprises an elastic and plastic regime upon application of tensile stress. Furthermore, with rare exceptions[17], the mechanical compliance is usually limited to one crystal facet, making it challenging to achieve mechanical flexibility in two dimensions, such as bending in two directions. Although PMA-III does not exhibit specific photochemical effects upon ultraviolet radiation, it shows good mechanical flexibility. To qualitatively assess its mechanical bending properties, simple bending tests were performed on this polymorph by using tweezers and a needle, which showed that application of force on (010) and (001) results in bending (Fig. 2c–h, Supplementary Movies 5 and 6). Specifically, when a small force is applied to the acicular crystal, it undergoes elastic bending along the (001) plane (Supplementary Fig. 11a). However, when the strain exceeds 0.7% (calculated based on the Euler–Bernoulli equation $\varepsilon_n = (t/2R)$[47,48], where $t$ is the thickness and $R$ is the radius of curvature), an irreversible plastic deformation occurs. Supplementary Fig. 11b illustrates that a typical single crystal of PMA-III has an elastic strain limit of 2.81% along the (010) plane. Once this threshold strain is exceeded, the crystal undergoes plastic deformation instead of breaking. Furthermore, by applying torque, a two-dimensional plastic twisting was observed (Fig. 2h). The basic mechanical properties of the three polymorphs were established by using nanoindentation (Supplementary Fig. 12, Supplementary Table 3), where load ($P$)–displacement ($h$) curves were obtained by indenting the crystals with a 5 mN load on their widest facet (Supplementary Fig. 12). The stress dissipation that inevitably accumulates in mechanically flexible crystals is dissipated via the intermolecular interactions, which may include interactions such as weak hydrogen bonding, halogen bonding, and van der Waals (vdW) interactions, and may be further assisted by reversible or irreversible movement of molecules. Although there are exceptions, a typical structure of a plastically bendable crystal presents an anisotropic packing arrangement where, for example, strong π−π columns are self-assembled across low-energy slip planes. Based on these inferences, analysis of the packing arrangement and intermolecular interactions was conducted in PMA to elucidate the bending mechanism (Fig. 3e).

In the crystals of the PMA polymorphs the molecules are assembled by a combination of weak interactions, mainly weak hydrogen bonding, π−π and vdW interactions (Fig. 3e). The adjacent PMA molecules in PMA-III are connected via C−H⋯N (3.465 Å, 2.541 Å, and 172.3°) and C−H⋯C (3.704 Å, 2.827 Å, and 157.5°) interactions to form hydrogen-bonded layers. Viewed along the $b$ axis, they are further arranged in columns via π−π (4.906 Å) and C−H⋯F (3.732 Å, 2.490 Å, 163.0°) interactions in the [001] direction. The hydrogen-bonded layers are arranged in a parallel fashion with each other along the $a$ axis and the $c$ axis, respectively. These columns further interact via vdW interactions to form a slip plane. Viewed from the $c$ axis, the PMA molecules form columns mainly via vdW interactions in the [010] direction, which further interact through vdW interactions along the $a$ axis to form a potential slip plane.

The current understanding of the flexibility of molecular crystals is that upon application of force, the molecules can reorient to allow compression of the crystal along the inner arc and expansion along the outer arc[33]. Once the strain exceeds the elastic threshold, stress dissipation can occur through slippage across slip planes, usually between molecular columns, leading to permanent plastic deformation (Supplementary Fig. 13). Based on the energies of intermolecular interactions in PMA-III, as visualized by energy frameworks, the interactions are rather anisotropic (Supplementary Figs. 14 and 15). The total interaction energy of the π−π columns along the $c$ axis is 72.5 kJ mol⁻¹. This value contrasts the total interaction energy between the columns, identified as a possible slip plane, at 11.8 kJ mol⁻¹. This is expected to facilitate the gliding of the columns passing each other under external mechanical force. The sharp Bragg diffraction peaks of a straight crystal of form PMA-III confirmed its high crystallinity (Supplementary Fig. 16). On the other hand, a crystal that has been bent on its (001) plane presented considerably sharp diffraction peaks of the (0$kl$) and ($hk$0) planes, but significantly broadened peaks of the ($h$0$l$) plane, which can be attributed to displacement of molecules from their original positions (Supplementary Figs. 16 and 17). Regrettably, like in some other cases[8,51], we were unable to obtain sufficiently good diffraction data to determine the structure of the crystal in the bent region. Infrared[52] and Raman spectroscopy, usually in their microversion due to the small crystal size, can be used to monitor the structure variation by collecting the spectrum in the different regions of bent crystals. However, in this work, no apparent peak shifting, broadening, or splitting was found in the µ-Raman spectrum of PMA-III on the outer or inner side of the bent region compared to the straight region (Supplementary Fig. 18). This goes along with earlier conclusions that infrared spectroscopy is more informative in resolving structural information in bent crystals compared to Raman spectroscopy[53]. More direct information on the structural changes in the bent region would require the application of synchrotron radiation[6], possibly in combination with theoretical calculations.

Considering the observed tendency of PMA-I crystals to elongate and split along their longest axis during the photoreaction, we deemed it necessary to relate these observations to their actuation performance. As shown in Supplementary Movie 7, irradiation of an acicular crystal of PMA-I and the ensuing photoreaction results in a considerable bending force that can propel a small steel ball. In this simple experiment, the actuated object acquires kinetic energy by conversion of light into mechanical work performed by the organic crystal. It is worth noting that the metal ball was thousands of times heavier than the crystal, and the crystal continued to bend even after the ball was pushed away, as expected by the increased yield with time[54–57]. To quantify the amount of useful work this material can perform by harnessing light and to evaluate its actuation, we analyzed its crystals' output force and work. Both acicular and prismatic crystals of PMA-I are photomechanically active, however, due to their different shapes they exhibit different photomechanical effects, and therefore we used different methods to assess their actuation performance. The results are listed in Table 1. The performance of the acicular single crystals in terms of their work density was estimated from the movement of the steel ball, as described above (Fig. 4d–h). The output force, $F_{out}$, was calculated with Eq. (1):

$$F_{out} = f_s^{max} = \mu m g \qquad (1)$$

where $f_s^{max}$ is the maximum static friction between the glass plate and the ball, $\mu$ is the static friction coefficient between the clean glass plate and the ball (usually 0.5–0.7; $\mu$ here was taken to be 0.6), and $m$ is the mass of the ball. From here, the output force density is:

$$F_{out}^v = \frac{F_{out}}{V} \qquad (2)$$

where $V$ is the volume of the crystal. The output work was calculated as the kinetic energy of the ball (Eq. (3)):

$$W_{out} = E = \frac{1}{2}mv^2 \qquad (3)$$

**Table 1 | Actuation performance of prismatic and acicular crystals of PMA-I[a]**

| | Prismatic | | Acicular | |
|---|---|---|---|---|
| | Output work density/J m$^{-3}$ | Force density/N m$^{-3}$ | Output work density/J m$^{-3}$ | Force density/N m$^{-3}$ |
| $n$[b] | 35 | | 35 | |
| Maximum | $2.85 \times 10^4$ | $6.59 \times 10^8$ | $3.94 \times 10^3$ | $8.98 \times 10^7$ |
| Minimum | $1.53 \times 10^3$ | $3.37 \times 10^7$ | $6.02 \times 10^1$ | $1.68 \times 10^6$ |
| Average | $9.84 \times 10^3$ | $2.34 \times 10^8$ | $1.23 \times 10^3$ | $2.94 \times 10^7$ |
| St. dev.[c] | $8.95 \times 10^3$ | $2.09 \times 10^8$ | $1.16 \times 10^3$ | $2.70 \times 10^7$ |

[a]The full data are provided in Supplementary Table 4.
[b]Number of samples.
[c]Standard deviation.

where $E$ is the kinetic energy, and $v$ is the speed of the ball. The output work density was calculated as:

$$W_{out}^v = \frac{W_{out}}{V} \qquad (4)$$

For the prismatic crystals of PMA-I, which tend to move and/or disintegrate during the reaction, however, we employed an alternative method. A crystal of PMA-I was placed between two glass plates (Fig. 4a–c). Being physically restrained, instead of exploding or jumping, during exposure to ultraviolet light the crystal expanded but macroscopically remained a single piece (Supplementary Movie 8). The output force and force densities were calculated by using Eqs. (1) and (2). The output work was measured from the displacement ($L$) of the two glass plates:

$$W_{out} = F_{out}L \qquad (5)$$

The relevance of the principal process of actuation performed by these and other photoactuating crystals lies in their potential application as crystalline machines for converting light into work. With the volume-normalized force and work densities at hand, we compared these two important parameters between the two crystal habits of the same polymorph (PMA-I) and also with those of other common actuators in the global materials property plot shown in Fig. 4i. It can be concluded that in terms of force density, PMA-I crystals of both habits outperform most of the known actuators, while in work density, they are competitive with many of the actuators. The prismatic crystals show a higher density of force and work, as anticipated from their more violent disintegration, where significant accumulated elastic energy is released within a short period of time. The acicular crystals are capable of generating force densities of $1.68–89.8 \times 10^6$ N m$^{-3}$, while the values for the prismatic crystals are $3.37–65.9 \times 10^7$ N m$^{-3}$. As discussed above, the photosalient effects are rapid disintegrative events that are related to the release of significant energy. On the other hand, photomechanical bending is a comparatively slower process, limited by the inefficient transduction of the molecular movements due to a photochemical reaction into macroscopic actuation. The photoconversion yields in this process are usually restricted by the localized conversion that occurs on the crystal surface, and the associated differential strain is slowly transferred to induce a bending moment.

The plot in Fig. 4i also highlights the good performance of PMA-I, and especially of the prismatic crystals of this material, when compared to other commonly used actuators that are based on multicomponent systems. The acicular crystals of PMA-I display work densities that range 0.06–3.94 kJ m$^{-3}$, values that are comparable to ceramic piezoelectric materials and voice coil actuators. Prismatic crystals are capable of even higher work densities of 1.5–28.5 kJ m$^{-3}$, that are competitive to thermal and diaphragm actuators, indicating their potential for high work output. The distinction between the two habits of the same material offers versatility with applications. While acicular crystals may be suitable for precision applications, in contrast, prismatic crystals could excel in scenarios where there is a demand for high force. The comparability of PMA-I crystals to diverse existing actuators suggests their capacity to compete and potentially surpass some of the current technologies, making the potential application of these materials a source of anticipation for future developments.

The results presented here demonstrate that multiple mechanical effects are possible even with a simple photochemical reaction of polymorphic crystalline solid. Having as many as three characterized polymorphs, the compound studied here was used to explore some of the factors that determine the type of mechanical effect and its relation to mechanical properties. Crystals of one of the forms of PMA bend, jump, and disintegrate upon exposure to ultraviolet radiation, and these effects can be associated with their crystal habit. On the contrary, crystals of another form roll under similar conditions. All these photomechanical effects are attributed to photochemical dimerization. Crystals of the third form are not reactive but can be elastically or plastically bent due to a combination of intermolecular interactions in their structure; they can be either bent or twisted, depending on the direction of the applied force. We have demonstrated that the photochemically reactive crystals are capable of harnessing light and converting it into work with force and power densities that supersede those of most of the common actuating devices, which favors these materials as lightweight, efficient micromachines. This work not only demonstrates the potential of using simple photochemical reactions in the solid state to actuate objects but also hints at the intricate interplay between the structure, morphology, and size in determining the type and magnitude of the related mechanical effects.

## Methods

### Material preparation and characterization

PMA was prepared by adding 3,5-bis(trifluoromethyl)phenylacetonitrile (2.53 g, 10 mmol) and 1-methyl-2-imidazolecarboxaldehyde (1.21 g, 11 mmol) in 60 mL ethanol. 15 mg NaOH was added to the solution. Yellow-green powder precipitated by stirring for three hours at 40 °C. The product was collected by filtration and washed with ethanol to obtain a yellow-green solid. To prepare the phrotoproduct, PMA-I was ground into microcrystalline powder and spread out in a very thin layer. The powder was exposed to 365 nm OLED ultraviolet lamp with power of 10 W for 2 h. The irradiated sample was recrystallized from acetonitrile. The acicular single crystals of PMA-I and PMA-II were obtained by layering ethanol on top of a dichloromethane solution containing PMA, followed by solvent diffusion at room temperature. Most of the obtained crystals were of form I (PMA-I) with a very small amount of form II (PMA-II). Crystals of PMA-III crystallize from the same solution if it is kept at 5 °C. The prismatic crystals of PMA-I were obtained by evaporative crystallization from a mixed solution in dichloromethane and ethanol. Single crystal of the photoproduct was prepared by slow evaporation from acetonitrile at 25 °C.

### UV induced photomechanical effects of PMA

For observation of the photomechanical effects, crystals of PMA-I and PMA-II were placed on a plastic base and irradiated by 365 nm OLED ultraviolet lamp. In case of directional radiation, when a certain crystal facet needed to be exposed to UV light, a crystal was glued to a metal needle. While observing the crystal face under a microscope, the crystal was exposed to ultraviolet light from a fixed source by rotating the metal needle.

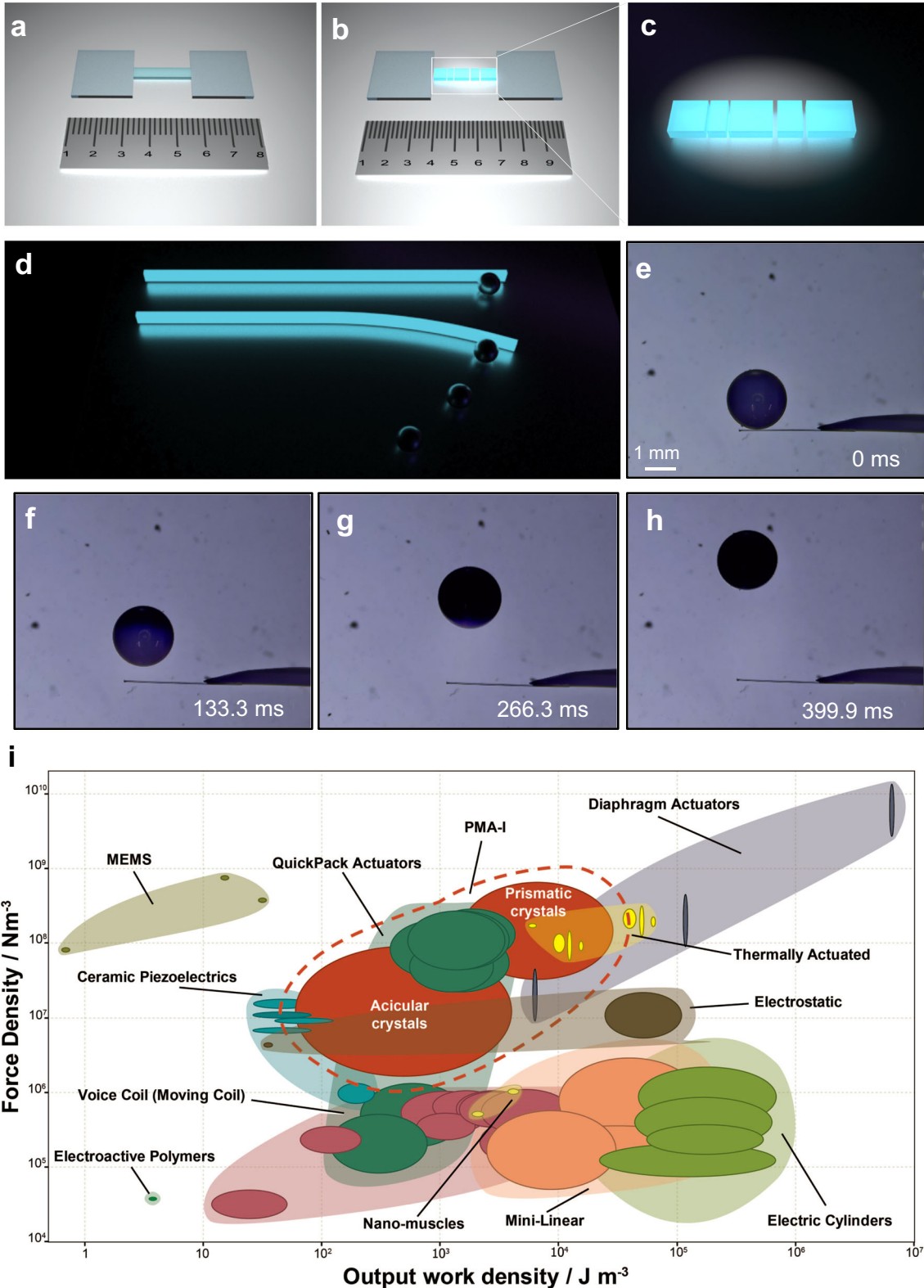

**Fig. 4 | Assessment of the actuating performance of PMA-I. a–c** Schematic diagram of the setup used to assess the actuation performance of blocky crystals that tend to disintegrate. **d** Schematic diagram of the experiment used to assess the actuation performance of acicular crystals. **e–h** Optical micrographs of an acicular crystal that actuates a steel ball. **i** Materials property plot that compares the force density and work density of PMA-I to other main actuator classes. The broken red line indicates the two sets of data obtained from crystals of PMA-I with different habit.

## Microscopy

The mechanical and photoresponsive behaviors were observed and recorded by using a SMZ745T microscope (Nikon). The high-speed recordings were obtained with a high-speed camera SH6-504 (SSZN), mounted on the microscope, at a speed of 2000 frames per second.

## Single crystal X-ray diffraction

A suitable single crystal was set on an XtaLAB Synergy-Custom system with HyPix detector (Rigaku). The data was collected at −160, −113 and 4 °C, while controlling the temperature with a Cryostream 800 cooler (Oxford Cryosystems). Olex2 was used to solve the structure by intrinsic phasing methods (SHELXT) and complete and refine the structure models using the full-matrix least-squares methods on $F^2$ (SHELXL)[58,59].

## Powder X-ray diffraction

Max-2500 X-ray diffractometer (Rigaku) was used to characterize the crystals in powder form. The tube voltage and the current were 40 kV and 100 mA, respectively. Data was collected from 2 to 35° at a scanning speed of 5° min$^{-1}$.

## μ-Raman spectroscopy

A single crystal of PMA-III that had been bent on the (001) facet was glued onto a glass slide. The Raman spectra were recorded by using a Raman microscope DXR (ThermoFisher Scientific), equipped with a 532 nm excitation laser operating at 2 mW, and by using a 50 μm slit.

## Nanoindentation

The nanomechanical tests were performed with KLA-G200 instrument. The wide facets of crystals of all three polymorphs were indented to a peak load of 5 mN with a loading/unloading rate of 0.25 mN s$^{-1}$ in all tests. The $P$−$h$ curves were analyzed by using the standard Oliver−Pharr method to extract the values of the Young's modulus ($E$) and hardness ($H$) of the wide plane of the crystals where the Poisson's ratio was taken to be 0.25.

## Scanning electron microscopy

To observe the mechanical bending and twisting of PMA-III, scanning electron microscopy was performed by using TM3000 microscope (Hitachi) with an accelerating voltage of 15 kV. The crystals were attached on a double-sided carbon tape.

## Fluorescence spectroscopy

A steady-state/transient fluorescence spectrometer FLS1000 (Edinburgh Instruments) was used to collect fluorescence spectral data and to determine the fluorescence lifetimes of single crystals. The excitation wavelength was 365 nm.

## Data availability

Crystal data of the PMA-I, PMA-II, PMA-III and photoproduct are available from the Cambridge Crystallographic Data Centre with reference numbers 2278400 and 2278402-2278404. All data are available from the corresponding authors upon request. Source data are provided with this paper.

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

## Acknowledgements

This work was supported by the National Natural Science Foundation of China (22378302 and 22178254) and Key Project of Tianjin (21JCZDJC00400), and a fund from New York University Abu Dhabi. This material is based upon works supported by Tamkeen under NYUAD RRC Grant No. CG011.

## Author contributions

J.L. and J.Z. performed the experiments. I.T. performed the performance analysis and prepared the figures of actuating performance. L.Li., P.N., S.W. and J.G. supervised the experiments. J.L., P.N., S.W. and J.G. conceived the project.

## Competing interests

The authors declare no competing interests.
