## [Peer Review File · Nature Communications]

Highly Efficient *in crystallo* Energy Transduction of Light to WorkREVIEWER COMMENTS

Reviewer #1 (Remarks to the Author):

The study contained in the manuscript entitled "Highly Efficient in crystallo Energy Transduction of Light to Work" is very interesting and I consider it as highly attractive to the materials chemistry and crystal engineering community. In this work, the authors have carefully carried out analyses of three polymorphs of a relatively simple molecule. Notably, each polymorph showed distinctive behavior under light irradiation or mechanical stress. Subsequently, the authors have carried out an in-depth description for the origin of the changes, based on the intermolecular interactions.

In addition, the authors quantified the actuator performance of one of the forms (PMA-I) and, interestingly, they found that it is competitive or surpasses other materials. Considering all the above, I believe this work is suitable for publication in this Journal.

I only have two minor suggestions, as enlisted below:

1. In figure 3, I could not find the labels c), d) or e)
2. Can the authors provide a powder X-ray diffraction spectrum of the photoproduct that is formed under irradiation of forms I and II? This would help to demonstrate that the phase of the product coincides with the recrystallized solid.

Overall, the work is sound, the methodology seems reproducible and the conclusions are well supported.

Reviewer #2 (Remarks to the Author):

In this manuscript, the authors report a confluence of different mechanical effects across three polymorphs of an organic crystalline cyanostyrene derivative that is capable of converting light into work with efficiency that surpasses other previously reported materials and devices. Very interesting results and good writing. There are some issues that need to be fixed. Overall, the results are of significant importance and impact and the paper is well-written. It is suitable for publication in Nature Communications after making the following revisions:

1. For PMA-II in Figure 1c, can the author provide a clearer picture, especially under the ultraviolet? For Figure 1h, can the author change the background color to make the crystal shape clearer and the scrolling effect more obvious? In addition, the author should indicate the direction of illumination in the picture.
2. Regarding the crystal of PMA - I bonding, the author claims that it will return to its original shape after long-term irradiation. Is there any real photo support for this phenomenon?
3. For the light bending video of PMA-I needle-shaped crystals, why is the light constantly flickering in

the picture? You can try fixing the light source to control the intensity and distance of the light.

4. For Supplementary Movie 3, Photoinduced bending of an acidic crystal of PMA-I shows that the crystal will continue to explore after bending, which should be mentioned in the relevant description of the main text, not just described as bending

5. Figure 3 lacks the image number annotation for c, d, e.

6. According to Supplementary Movie 7, we have learned that needle-shaped PMA-I crystals have good workability, but we are curious why the author inverted the image in supporting video 8 in Figures 4e-h. We believe that the two should be consistent, and we ask the author to provide an explanation.

Reviewer #3 (Remarks to the Author):

In this manuscript, which was titled "Highly Efficient in crystallo Energy Transduction of Light to Work" the authors obtained three polymorphs of imidazolephenyl acrylonitrile crystal. Interestingly, multiple mechanical effects of crystals are found and the interplay between the structure, morphology, and size in determining the type and magnitude of the related mechanical effects were studied. Furthermore, the potential applications of these crystals were also demonstrated, respectively.

The results and conclusions in this article are of interest and have potential impact on chemistry and material science of crystal. I would like to recommend for publication in "nature communications" after the authors adequately address some points as follows:

1. In the manuscript, the author described that when different facet of acicular crystals of PMA-I were irradiated by ultraviolet, the crystal bent away from the light source and a model was provided in Supplementary Figure 5, please described how to control the facet of the acicular crystal face to the light in the methodology. The author also found that the bent crystals recover their straight state upon prolonged irradiation (line 105 page 5) but in the movie 3 this phenomenon was not included. Please add the movie of this phenomenon.

2. In the manuscript, the author described that three new signals appeared in the ¹HNMR of product at 6.05 ppm, 6.84 ppm and 7.83ppm, respectively. Actually, there one signal ascribed to the double bond of reactant disappeared and one new signal ascribed to the cyclobutane ring of product appeared, all other four signals ascribed to the aromatic rings have shift apparently after irradiation. So please revised the corresponding description in the manuscript.

3. There are some impurities in the ¹HNMR of PMA (Supplementary Figure 2) and TMC (Supplementary Figure 8). Both compounds should be purified and clear ¹HNMR spectra should be provided.

4. The high-quality of Figure 1 c and h are needed in high-level work, therefore, the corresponding revisions are needed.

5. There are some mistakes in the manuscript, and corresponding revision are nedded:

1) It should be "1-methyl-1H-imidazole-2-carbaldehyde" in line 66 page 3.

2) The product of photocycloaddion should be cyclobutane (line 142 page 6).

3) In Figure 3 page 8 the icon of c-e was missing.

Response to the comments from Reviewer #1:

Comment: *The study contained in the manuscript entitled "Highly Efficient in crystallo Energy Transduction of Light to Work" is very interesting and I consider it as highly attractive to the materials chemistry and crystal engineering community. In this work, the authors have carefully carried out analyses of three polymorphs of a relatively simple molecule. Notably, each polymorph showed distinctive behavior under light irradiation or mechanical stress. Subsequently, the authors have carried out an in-depth description for the origin of the changes, based on the intermolecular interactions.*

In addition, the authors quantified the actuator performance of one of the forms (PMA-I) and, interestingly, they found that it is competitive or surpasses other materials. Considering all the above, I believe this work is suitable for publication in this Journal.

I only have two minor suggestions, as enlisted below:

Comment: *1. In figure 3, I could not find the labels c), d) or e).*

Response: We thank the Reviewer for the comment. We have redrawn the Figure 3 and added the labels.

Revised figure:

Revised Figure 3. Crystal structures of the three forms of PMA. (a) Intra- π -dimer structure PMA-I. (b) Intra- π -dimer structure PMA-II. (c) Inter- π -dimer packing of PMA-I. (d) Inter- π -dimer packing of PMA-II (e) Molecular packing of PMA-III. The distances of the most relevant intermolecular interactions are highlighted in red.

Comment: 2. Can the authors provide a powder X-ray diffraction spectrum of the photoproduct that is formed under irradiation of forms I and II? This would help to demonstrate that the phase of the product coincides with the recrystallized solid.

Response: We thank the Reviewer for this suggestion. We have provided the powder X-ray spectrum of the photoproduct of both forms and the recrystallized solid in the revised Supporting Information. It showed that the productions of irradiation of PAM-I and PMA-II and the product obtained after recrystallization had different X-ray diffraction patterns, revealing that their different

New Supplementary Figure 9. PXRD patterns of the products obtained by irradiation of PAM-I and PMA-II, and after recrystallization.

Added text: “The photoproducts of PMA-I and PMA-II had different X-ray diffraction patterns that were also different from that of the recrystallized product, and therefore they were different phases (Supplementary Figure 9).” (Page 7)

Response to the comments from Reviewer #2:

Comment: *In this manuscript, the authors report a confluence of different mechanical effects across three polymorphs of an organic crystalline cyanostyrene derivative that is capable of converting light into work with efficiency that surpasses other previously reported materials and devices. Very interesting results and good writing. There are some issues that need to be fixed. Overall, the results are of significant importance and impact and the paper is well-written. It is suitable for publication in Nature Communications after making the following revisions:*

Comment: *1. For PMA-II in Figure 1c, can the author provide a clearer picture, especially under the ultraviolet? For Figure 1h, can the author change the background color to make the crystal shape clearer and the scrolling effect more obvious? In addition, the author should indicate the direction of illumination in the picture.*

Response: We thank the Reviewer for the comment. We have improved the quality of Figures 1c and 1h and also added the direction of illumination in Figure 1h.

Revised Figure 1. Appearance and mechanical effects of crystals of the PMA polymorphs. (a) Chemical formula of PMA. (b–d) Images of crystals of the three polymorphs of PMA under visible and ultraviolet light: PMA-I (b), PMA-II (c), and PMA-III (d). (e–h) Photomechanical effects of PMA-I (e–g) and PMA-II (h) exposed to ultraviolet light.

Comment: 2. Regarding the crystal of PMA - I bonding, the author claims that it will return to its original shape after long-term irradiation. Is there any real photo support for this phenomenon?

Response: We thank the Reviewer for the comment and the careful readings. We reinvestigated the movie which recorded the phenomenon. Unfortunately, we found that the “return to original shape” conclusion was not always reproducible. The crystal flipped during the radiation process which resulted in its opposite face being exposed to light and eventually straighten the crystal (Figure R1c-f). The constant radiation usually damages the crystal once exceeding the stress limit (Figure R1a-c). Therefore, we apologize for the misinterpretation, and we concluded that it is better to remove the related statement from the revised manuscript.

Figure R1 (for review purposes only). Photomechanical effects of acicular PMA-I crystals.

Comment: 3. For the light bending video of PMA-I needle-shaped crystals, why is the light constantly flickering in the picture? You can try fixing the light source to control the intensity and distance of the light.

Response: We thank the Reviewer for pointing this out. The flickering effect is a result of the fact that the light source was hand-held, and was not fixed during the experiment. Following the reviewer’s suggestion, Supplementary Movie 3 was replaced with a movie with a fixed light source, and Figure 1 was replaced as well, as they are frames from the Supplementary Movie 3.

Comment: 4. For Supplementary Movie 3, Photoinduced bending of an acidic crystal of PMA-I shows that the crystal will continue to explore after bending, which should be mentioned in the relevant description of the main text, not just described as bending.

Response: We thank the Reviewer for pointing this out. The following text has been added (Page 5, line 103) to the revised version of the manuscript:

Added text: “When the bending strain exceeded the limit, they were broken and accompanied by salient behaviors.”

Comment: 5. Figure 3 lacks the image number annotation for c, d, e.

Response: We thank the Reviewer for this remark. We have redrawn the Figure 3 and added the panel numbers.

New Figure 3. Crystal structures of the three forms of PMA. (a) Intra- π -dimer structure PMA-I. (b) Intra- π -dimer structure PMA-II. (c) Inter- π -dimer packing of PMA-I. (d) Inter- π -dimer packing of PMA-II (e) Molecular packing of PMA-III. The distances of the most relevant intermolecular interactions are highlighted in red.

Comment: 6. According to Supplementary Movie 7, we have learned that needle-shaped PMA -I crystals have good workability, but we are curious why the author inverted the image in supporting video 8 in Figures 4e-h. We believe that the two should be consistent, and we ask the author to provide an explanation.

Response: We thank the Reviewer for pointing this out. We had accidentally inverted the picture when we took the images from the video to make the Figure. We have now redrawn this figure, and the new version is included in the revised manuscript.

Revised Figure 4. Assessment of the actuating performance of PMA-I. (a–c) Schematic diagram of the setup used to assess the actuation performance of blocky crystals that tend to disintegrate. (d) Schematic diagram of the experiment used to assess the actuation performance of acicular crystals. (e–h) Optical micrographs of an acicular crystal that actuates a steel ball. (i) Materials property plot that compares the force density and work density of PMA-I to other main actuator classes. The broken red line indicates the two sets of data obtained from crystals of PMA-I with different habit.

Response to the comments from Reviewer #3:

Comment: *In this manuscript, which was titled “Highly Efficient in crystallo Energy Transduction of Light to Work” the authors obtained three polymorphs of imidazolephenyl acrylonitrile crystal. Interestingly, multiple mechanical effects of crystals are found and the interplay between the structure, morphology, and size in determining the type and magnitude of the related mechanical effects were studied. Furthermore, the potential applications of these crystals were also demonstrated, respectively. The results and conclusions in this article are of interest and have potential impact on chemistry and material science of crystal. I would like to recommend for publication in “nature communications” after the authors adequately address some points as follows:*

Comment: 1. *In the manuscript, the author described that when different facet of acicular crystals of PMA-I were irradiated by ultraviolet, the crystal bent away from the light source and a model was provided in Supplementary Figure 5, please described how to control the facet of the acicular crystal face to the light in the methodology. The author also found that the bent crystals recover their straight state upon prolonged irradiation (line 105 page 5) but in the movie 3 this phenomenon was not included. Please add the movie of this phenomenon.*

Response: We thank the Reviewer for the comment. We have added the related description of control method in the revised manuscript (Page 13, line 332). We reinvestigated the movie which recorded the phenomenon. Unfortunately, we found that the “return to original shape” phenomenon was not a solid fact. The crystal flipped during the radiation process which resulted in the counter face was irradiated and eventually straighten the crystal (Figure R1c-f). The constant radiation usually damages the crystal once exceeding the stress limit (Figure R1a-c). Therefore, we apologize for the mis-interpretation and deleted the related statement in the revised manuscript.

Added text: “In case of directional radiation, when a certain crystal facet needed to be exposed to UV light, a crystal was glued to a metal needle. While observing the crystal face under a microscope, the crystal was exposed to ultraviolet light from a fixed source by rotating the metal needle.”

Figure R1 (for review purposes only). Photomechanical effects of acicular PMA-I crystals.

Comment: 2. In the manuscript, the author described that three new signals appeared in the ^1H NMR of product at 6.05 ppm, 6.84 ppm and 7.83ppm, respectively. Actually, there one signal ascribed to the double bond of reactant disappeared and one new signal ascribed to the cyclobutane ring of product appeared, all other four signals ascribed to the aromatic rings have shift apparently after irradiation. So please revised the corresponding description in the manuscript.

Response: We thank the Reviewer for this comment. We have revised the related text (Page 6, line 140) in the new manuscript.

Added text: "Compared to the ^1H NMR spectrum of the sample before irradiation, the signal from the proton at the double bond (8.07 ppm) disappeared and a new signal at 6.05 ppm appeared, showing the generation of the cyclobutane ring. The three signals from the aromatic protons shifted from 8.43, 7.51 and 7.30 ppm to 7.83, 7.26 and 6.84 ppm, respectively and one signal at 8.15 ppm had no shift. And the signals from the -methyl shifted from 3.94 to 3.91 ppm (Supplementary Figure 2 and 8)."

Comment: 3. There are some impurities in the ^1H NMR of PMA (Supplementary Figure 2) and TMC (Supplementary Figure 8). Both compounds should be purified and clear ^1H NMR spectra should be provided.

Response: We thank the Reviewer for pointing this out. The synthetic PMA powder was washed three times with ethanol. The crystals of the reactive polymorphs PMA were irradiated for 1 hour under ultraviolet light (maximum at 365 nm) to obtain the TMC crystals, which further recrystallized in acetonitrile for twice. The clear ^1H NMR spectra have be provided below and in the revised the supporting information.

New Supplementary Figure 2. ^1H NMR spectrum of PMA in DMSO-d_6 (400 MHz).

New Supplementary Figure 8. ^1H NMR spectra of TMC in DMSO-d_6 (400 MHz).

Comment: 4. *The high-quality of Figure 1 c and h are needed in high-level work, therefore, the corresponding revisions are needed.*

Response: We thank the Reviewer for the comment. In the revised version, we have provided clearer pictures.

Revised Figure 1. Appearance and mechanical effects of crystals of the PMA polymorphs. (a) Chemical formula of PMA. (b–d) Images of crystals of the three polymorphs of PMA under visible and ultraviolet light: PMA-I (b), PMA-II (c), and PMA-III (d). (e–h) Photomechanical effects of PMA-I (e–g) and PMA-II (h) exposed to ultraviolet light.

Comment: 5. There are some mistakes in the manuscript, and corresponding revision are needed:

- 1) It should be “1-methyl-1H-imidazole-2-carbaldehyde” in line 66 page 3.
- 2) The product of photocycloaddition should be cyclobutane (line 142 page 6).
- 3) In Figure 3 page 8 the icon of c-e was missing.

Response: We thank the Reviewer for the comment. We have corrected these mistakes in the revised manuscript. The revised version of Figure 3 was provided below.

Revised Figure 3. Crystal structures of the three forms of PMA. (a) Intra- π -dimer structure PMA-I. (b) Intra- π -dimer structure PMA-II. (c) Inter- π -dimer packing of PMA-I. (d) Inter- π -dimer packing of PMA-II (e) Molecular packing of PMA-III. The distances of the most relevant intermolecular interactions are highlighted in red.

REVIEWERS' COMMENTS

Reviewer #1 (Remarks to the Author):

After reviewing the additions and corrections that are now included in the revised version of the manuscript, I find it very well written and scientifically relevant for the materials science community. Therefore, I consider that this version is suitable for publication in Nature Communications

Reviewer #2 (Remarks to the Author):

The authors have addressed all questions. I recommend an acceptance.

Reviewer #3 (Remarks to the Author):

The author have revised the manuscript entitled "Highly Efficient in crystallo Energy Transduction of Light to Work" detailedly according to all the reviewers' comments. Considering the interesting results and good writting of this work, I think it is suitable to publish in "nature communications" without futher revisions.